# Diverse Members of the Phylum *Armatimonadota* Promote the Growth of Aquatic Plants, Duckweeds

**DOI:** 10.3390/ijms26199824

**Published:** 2025-10-09

**Authors:** Tomoki Iwashita, Ayaka Makino, Ryosuke Nakai, Yasuko Yoneda, Yoichi Kamagata, Tadashi Toyama, Kazuhiro Mori, Yasuhiro Tanaka, Hideyuki Tamaki

**Affiliations:** 1Biomanufacturing Process Research Center, National Institute of Advanced Industrial Science and Technology (AIST), Tsukuba 305-8560, Ibaraki, Japan; iwashita.t@aist.go.jp (T.I.);; 2Research Institute of Energy, Environment and Geology, Hokkaido Research Organization (HRO), Sapporo 060-0819, Hokkaido, Japan; 3Biomanufacturing Process Research Center, National Institute of Advanced Industrial Science and Technology (AIST), Sapporo 062-8517, Hokkaido, Japan; 4Graduate Faculty of Interdisciplinary Research, University of Yamanashi, Kofu 400-8510, Yamanashi, Japan; 5Graduate School of Life and Environmental Sciences, University of Yamanashi, Kofu 400-8510, Yamanashi, Japan

**Keywords:** duckweed, plant growth-promoting bacteria, *Armatimonadota*, frond

## Abstract

Duckweeds are small, fast-growing aquatic plants with high starch and protein content, making them promising candidates for next-generation plant biomass resources. Despite their importance, little is known about their interactions with microorganisms, particularly plant growth-promoting bacteria (PGPB), which play key roles in enhancing plant productivity. In this study, we report the plant growth-promoting effects of strain LA-C6, a member of the phylum *Armatimonadota*, isolated from duckweed fronds. Based on 16S rRNA gene analysis, this strain represents a novel genus-level lineage, and is referred to as *Fimbriimonadaceae* bacterium strain LA-C6. In axenic co-culture experiments, strain LA-C6 promoted duckweed growth, increasing the frond proliferation of four duckweed species (*Lemna minor*, *Lemna aequinoctialis*, *Spirodela polyrhiza*, and *Landoltia punctata*) by 1.8- to 4.0-fold compared with uninoculated controls. Importantly, three other phylogenetically distinct *Armatimonadota* species also exhibited significant plant growth-promoting effects on *L. minor*, increasing frond number by up to 2.3-fold and dry weight by up to 2.4-fold. This finding highlights the broader potential of diverse *Armatimonadota* members as PGP bacteria. A survey of the IMNGS database showed that strain LA-C6 and other *Armatimonadota* species are widely distributed across diverse plant-associated environments. Biochemical assays and gene prediction analyses revealed that strain LA-C6 produces indole-3-acetic acid (IAA) as a representative PGP trait, whereas no additional PGP-associated traits were detected. These results suggest that diverse bacterial lineages within the phylum *Armatimonadota* exert growth-promoting effects on aquatic plants, potentially through yet-to-be-identified mechanisms.

## 1. Introduction

The growth, health, and productivity of plants are supported and enhanced by beneficial symbiotic microorganisms known as plant growth-promoting bacteria (PGPB) [1,2]. PGPB have been extensively studied in agricultural and forestry research fields, as their inoculation can significantly promote plant growth and biomass [3,4,5,6]. Numerous PGPB strains have been isolated from terrestrial plants, and their diverse mechanisms of plant growth promotion, including phytohormone production, nutrient supply, and pathogen suppression, have been well characterized [7,8]. However, previous studies have shown a strong phylogenetic bias, with most well-known PGPB strains belonging to just five major phyla: *Pseudomonadota*, *Actinomycetota*, *Bacteroidota*, *Bacillota*, and *Cyanobacteriota* [9].

Duckweeds (subfamily *Lemnoideae*) are small, free-floating aquatic plants that are widely distributed in freshwater environments worldwide. Taxonomically, the subfamily *Lemnoideae* comprises five genera (*Spirodela*, *Lemna*, *Landoltia*, *Wolffia*, and *Wolffiella*) including 37 species, which exhibit a wide range of morphological characteristics. The genera *Spirodela*, *Lemna*, and *Landoltia* typically consist of two parts: fronds (a fusion of the leaf and stem) and roots. In contrast, *Wolffia* and *Wolffiella* lack roots and consist solely of fronds. Duckweeds are characterized by high starch and protein content, making them promising resources for energy production, animal feed, and even human food sources [10,11,12,13]. To enhance biomass productivity and growth potential, the application of PGPB to duckweeds has been explored. Indeed, several beneficial strains have recently been obtained [14,15,16,17,18,19]. However, the number of PGPB strains recovered from duckweeds remains far lower than those obtained from terrestrial plants, and their phylogenetic diversity appears similarly restricted to the five major phyla commonly observed in terrestrial systems. Recently, we isolated two bacterial strains belonging to the phylum *Acidobacteriota*, known as a rarely cultivated bacterial group, and demonstrated their ability to promote growth of various duckweed species [20]. These results suggest that duckweeds may harbor unique PGPB whose ecological functions remain largely unknown.

In a previous study, we successfully isolated strain LA-C6, a taxonomically novel member of the rarely cultured phylum *Armatimonadota*, from a duckweed sample [21]. Although *Armatimonadota* bacteria have been reported in a wide variety of natural environments such as soil, freshwater, and hot springs [22,23,24,25,26,27], their functions and ecological roles remain largely unknown, and no studies to date have reported a direct association with plants. Furthermore, our findings indicate that *Armatimonadota* bacteria are distributed in various duckweed species, accounting for approximately 1–5% of the microbial community [21,28], suggesting that they may be involved in duckweed growth.

In this study, we evaluated the plant growth-promoting effects and colonization abilities of strain LA-C6 across several duckweed species. In addition, three previously isolated *Armatimonadota* species (*Armatimonas rosea*, *Fimbriimonas ginsengisoli*, and *Capsulimonas corticalis*), each belonging to lineages distinct from strain LA-C6, were assessed for their plant growth-promoting effects on *Lemna minor*. To further characterize their potential roles, we investigated the plant growth-promoting traits and environmental distribution of these *Armatimonadota* species through phenotypic assays, genomic analyses, and 16S rRNA gene-based surveys.

## 2. Results

### 2.1. Plant Growth-Promoting Effect on Duckweeds by the Novel Armatimonadota Strain LA-C6

In a previous study, we isolated bacterial strain LA-C6 from fronds of *Lemna aequinoctialis* grown in artificial ponds, and its partial 16S rRNA gene sequence (accession number LC523985) indicated that this strain is a new member of the phylum *Armatimonadota* [21]. To further determine the phylogenetic placement of this strain, the full-length 16S rRNA gene (1503 bp), derived from genome sequences described below (accession number AP038796), was sequenced and subjected to homology search against the EzBioCloud database in this study. Strain LA-C6 was most closely related to *Fimbriimonas ginsengisoli* strain Gsoil 348^T^, isolated from the ginseng field soil (92.3% sequence similarity), while its homology to other *Armatimonadota* species was 82.9% with *A. rosea* and 81.3% with *C. corticalis*. The phylogenetic tree based on the nearly full-length 16S rRNA gene sequences revealed that strain LA-C6 belongs to the order *Fimbriimonadales* with robust bootstrap support (100%) (Figure 1). These results clearly indicate that this strain represents a phylogenetically novel lineage at least at the genus level, therefore it is referred to as *Fimbriimonadaceae* bacterium strain LA-C6.

To evaluate the plant growth-promoting effects on duckweeds, six aseptic duckweed species were selected, and two independent experiments were conducted (Appendix A). First, each duckweed plant was soaked and incubated in modified Hoagland medium containing strain LA-C6 cultured cells (cell suspension method), and the number of fronds was counted (Appendix A). Strain LA-C6 significantly promoted growth, increasing the number of fronds by 4.0-, 3.6-, 1.8-, and 3.0-fold in *L. minor*, *L. aequinoctialis*, *Spirodela polyrhiza*, and *Landoltia punctata*, respectively, compared with non-inoculated controls (Figure 2), while no plant growth-promoting (PGP) effects were found in *Wolffia* species. These results indicate that the PGP effects of strain LA-C6 are not specific to *L. aequinoctialis,* the isolation source of the strain, but are also observed in multiple other duckweed species.

We further evaluated the PGP effects of strain LA-C6 in an alternative system involving LA-C6-attached duckweeds (pre-attached method) (Appendix A). Aseptic duckweeds were co-incubated with strain LA-C6 for three days, after which the colonized duckweeds were transferred to fresh modified Hoagland medium and cultivated for 14 days (Appendix A). Strain LA-C6 significantly enhanced the growth of *S. polyrhiza*, *L. minor*, and *L. aequinoctialis* by up to 2.6-fold (Figure 3), suggesting that bacterial attachment to the plant surface alone is sufficient to promote duckweed growth.

### 2.2. Plant Growth-Promoting Effect of Various Armatimonadota Species

Since strain LA-C6 enhanced duckweed growth, we further evaluated the PGP effects of three additional *Armatimonadota* species: *A. rosea*, *C. corticalis*, and *F. ginsengisoli*. These species represent broad phylogenetic diversity within the phylum *Armatimonadota: A. rosea* (order *Armatimonadales*), *C. corticalis* (order *Capsulimonadales*), and *F. ginsengisoli* (order *Fimbriimonadales*) (Table 1). All *Armatimonadota* strains, including strain LA-C6, increased the number of fronds by 2.1- to 2.5-fold and dry weight by 1.9- to 2.1-fold compared with non-inoculated controls (Figure 4). As a positive control, *Acinetobacter calcoaceticus* strain P23, a well-known PGPB for duckweed, enhanced both frond number and dry weight by 1.8-fold. These results suggest that phylogenetically diverse *Armatimonadota* species exhibit PGP activity comparable to those of representative PGPB for duckweed.

### 2.3. Colonization of Duckweed Surfaces by Strain LA-C6

The colonization of strain LA-C6 on duckweed surfaces was evaluated on *L. minor* and *L. aequinoctialis* using qPCR. In *L. minor*, strain LA-C6 colonized the plant surface at approximately 10^4^–10^5^ cells/mg plant tissue after 3 days of co-cultivation (Figure 5A). In *L. aequinoctialis*, the colonization by strain LA-C6 was observed at approximately 10^3^–10^4^ cells/mg plant tissue after 7 days of co-cultivation (Figure 5B). Fluorescence microscopy was further used to examine colonization of strain LA-C6 on duckweed surfaces. As shown in Figure 6, live LA-C6 cells were detected on both fronds and roots of *L. minor* and *L. aequinoctialis*, indicating that this strain can attach to multiple parts of the duckweed surface.

### 2.4. Assays for Bacterial Plant Growth-Promoting Traits

Typical plant growth-promoting traits, such as siderophore production, phosphate solubilization, and IAA production, were examined for strain LA-C6. Strain LA-C6 was positive for IAA production, but negative for phosphate solubilization and siderophore production. Other *Armatimonadota* strains were also evaluated for these traits. Phosphate solubilization activity was observed in *F. ginsengisoli* and *C. corticalis*. IAA production was detected in *A. rosea* and *C. corticalis* when grown in R2A medium supplemented with 0.5% (*w/v*) L-tryptophan. No siderophore production was observed in any of the *Armatimonadota* strains examined.

### 2.5. PGP-Associated Genes in the Genome of Strain LA-C6

Using the Revio sequencing system, we successfully obtained the complete genome sequence of the LA-C6 strain (Genbank accession number AP038796). To confirm the PGP potential of strain LA-C6, genes associated with typical PGP traits were predicted through both automated gene annotation and additional manual curation. Genes involved in siderophore synthesis (e.g., *ent*, *pvd*) were not identified, suggesting that strain LA-C6 is unable to produce typical siderophores. Phosphate solubilization is generally mediated by the secretion of organic acids. In particular, gluconic acid and α-ketogluconic acid are widely regarded as key compounds responsible for phosphate solubilization [29]. However, genes involved in the metabolism of gluconic acid (e.g., *gcd*, *pqq*) and α-ketogluconic acid (*gad*) were not detected in strain LA-C6.

IAA biosynthesis is known to occur via several pathways, including the indole-3-acetamide pathway, indole-3-acetonitrile pathway, indole-3-pyruvate pathway, and tryptamine pathway [30]. Among these pathways, partial gene sets such as tryptamine oxidase (EC1.4.3.4) and indole-3-acetamide amidohydrolase (EC3.5.1.4) were predicted in the genome of strain LA-C6. However, none of these pathways were found to be complete (Appendix A). Given that IAA production by strain LA-C6 was experimentally confirmed, further genome searches were conducted to identify missing key genes, particularly those related to tryptophan 2-monooxygenase. A candidate gene containing a domain classified within the tryptophan 2-monooxygenase family was identified using BLASTp and Pfam analyses.

In addition, genes related to other PGP traits that were not experimentally assessed were also evaluated. Consequently, nitrogen fixation genes (*nifHDK*, *vnfHDK*, and *anfHDK*) [31], phosphonate metabolism genes (*phn* gene cluster) [32], acetoin/2,3-butanediol synthesis genes (*budA*, *B*, *C*) [33], nitric oxide production gene (*nirK*) [34], and ACC deaminase gene (*acdS*) [35] were not detected in the genome of strain LA-C6. These findings suggest that, apart from IAA production, strain LA-C6 likely lacks the genetic basis for other typical PGP traits.

### 2.6. Environmental Distribution of Strain LA-C6 and Other Armatimonadota Strains

IMNGS database search was conducted to investigate the relative abundance of strain LA-C6 across three environmental categories: plant-associated, soil, and water. Strain LA-C6 was broadly distributed, particularly in plant-associated samples (> 3% positive samples, including roots, plant, and rhizosphere) and soil samples (> 3% positive samples, including rice paddy, wetland, peat, and soil) (Figure 7). Among other *Armatimonadota* species, *F. ginsengisoli* was predominantly found in plant-associated samples, especially in the rhizosphere. *C. corticalis* was frequently detected in both plant-associated samples (roots, rhizosphere, and leaf litter) and soil samples (peat and soil). In contrast, *A. rosea* was mainly found in plant-associated samples (roots) and water samples (freshwater, aquatic, lake water, groundwater, estuary, and riverine). These findings indicate that all *Armatimonadota* species-related members, including strain LA-C6, are widely distributed in diverse plant-associated habitats.

## 3. Discussion

Duckweeds have attracted attention as valuable biomass resources due to their high protein and starch content [10,11,12,13]. Because of their ease of handling, the creation of sterile strains, and fast growth rates, duckweeds have been used as model plants for research on plant–microbe interactions, particularly with plant growth-promoting bacteria (PGPB) [18,36,37,38]. Recently, several PGPB have been isolated from duckweeds, including strains belonging to phylogenetic groups distinct from those reported in terrestrial plants [17,19,20]. This suggests that aquatic environments may harbor PGPB that have been overlooked. Our previous studies showed that bacteria within the phylum *Armatimonadota*, known as a rarely cultivated bacterial group, are distributed across a wide range of duckweed species [21,28]. While we successfully isolated several strains, their symbiotic relationships with plants remain unknown. The present study is the first to clearly demonstrate the plant growth-promoting effect of a newly isolated *Armatimonadota* strain on duckweeds, and to reveal the phylogenetic breadth of such PGP strains within the phylum *Armatimonadota*.

A taxonomically novel isolate of the phylum *Armatimonadota*, strain LA-C6, was found to enhance the growth of various duckweed species using two evaluation methods: the cell suspension method (Appendix A) and the pre-attached method (Appendix A). In the cell suspension method, strain LA-C6 significantly promoted the growth of *L. minor*, *L. aequinoctialis*, and *S. polyrhiza* by 4.0-, 3.6-, and 1.8-fold, respectively, compared with non-inoculated controls. In the pre-attached method, the strain enhanced the growth of *L. minor* and *L. aequinoctialis* by 2.2- and 2.6-fold, respectively. The PGP effect of strain LA-C6 was comparable to that of previously reported PGPB for duckweeds. For example, strain P23, the most well-characterized duckweed-associated PGPB, was reported to increase frond numbers by 3.8-, 2.9-, and 1.4-fold in *L. minor*, *L. aequinoctialis*, and *S. polyrhiza*, respectively, after 7 days of co-cultivation using the cell suspension method [39]. In the pre-attached method, strain P23 also promoted the growth of *L. minor*, with a 1.8-fold increase in frond number compared with non-inoculated controls. Our previous study also demonstrated that PGPB affiliated with *Acidobacteriota* enhanced the growth of *L. minor*, *L. aequinoctialis*, and *S. polyrhiza* by 2.4-, 3.1-, and 2.2-fold, respectively, after 14 days of co-cultivation in the cell suspension method [20]. Ishizawa et al. [15] also reported that five bacterial strains promoted the growth of *L. minor* by up to 1.2-fold compared with non-inoculated controls after 7 days in the pre-attached method. These findings suggest that strain LA-C6 exhibits PGP effects equal to or greater than those of previously reported duckweed-associated PGPB. Colonization assays showed that strain LA-C6 attached to both the fronds and roots of *L. minor* and *L. aequinoctialis*. Due to high substrate diffusion in aquatic environments, stable colonization on duckweed surfaces is considered crucial for effective plant growth promotion [20]. These results further support the fact that strain LA-C6 is particularly effective in promoting the growth of *L. minor* and *L. aequinoctialis*.

We further demonstrated that various *Armatimonadota* species enhanced the growth of *L. minor* (frond number: 1.6- to 2.3-fold; dry weight: 1.8- to 2.4-fold). This is the first report indicating that *Armatimonadota* bacteria exhibit growth-promoting effects on plants. In this study, we also evaluated the PGP effects using duckweeds pre-attached with each strain in sterile modified Hoagland medium and found that all tested strains showed significant PGP potential. Although these strains were originally isolated from plant-associated environments [40,41,42], the associated plant species and environmental conditions varied. This suggests that *Armatimonadota* bacteria may form symbiotic relationships with a broad range of plant species across diverse habitats. The discovery of new PGPB in three different orders, *Armatimonadales*, *Capsulimonadales*, and *Fimbriimonadales*, further suggests that plant growth-promoting capabilities are widely distributed across the phylum *Armatimonadota*.

Activity assays and genome-based analyses for PGP traits confirmed that strain LA-C6 is capable of producing IAA. However, previous studies showed that exogenous IAA had no significant positive effect on duckweed growth [20,43,44], suggesting that IAA production is unlikely to have contributed to the PGP ability of strain LA-C6. Genomic analyses revealed that strain LA-C6 lacks complete gene sets associated with typical PGP traits, including phosphate solubilization, siderophore production, gluconic acid and α-ketogluconic acid metabolism, and nitrogen fixation, except for IAA production. Previous studies also demonstrated that typical PGP factors for terrestrial plants (e.g., phosphate solubilization, siderophore production, and HCN production) do not affect duckweed growth [44]. Interestingly, *Acidobacteriota* PGPB isolated from duckweeds also exhibited IAA production, as with strain LA-C6, whereas no other known typical PGP factors were detected by activity assays and genomic analyses [20]. Therefore, alternative mechanisms may be involved, such as the modulation of duckweed stress tolerance (e.g., oxidative stress, light intensity, and nutrient deficiency), or the supply of unknown substances that enhance metabolism and biomass yield. However, these possibilities remain to be further clarified. These findings suggest that strain LA-C6 may rely on novel PGP factors that are distinct from typical ones. Two plausible explanations can be considered: (i) strain LA-C6 may be specifically adapted to exert PGP effects in aquatic rather than terrestrial environments; and (ii) it belongs to a unique phylogenetic lineage, the phylum *Armatimonadota*, which is distinct from those of previously reported PGPB.

Four *Armatimonadota* species, including strain LA-C6, were frequently detected in plant-associated samples based on IMNGS analyses. Notably, *Armatimonadota* bacteria have been reported to be distributed around aquatic plants in previous studies [21,45,46,47]. Considering the observed PGP effects on duckweeds, this phylum may play a key role in promoting the growth of aquatic plants. In addition, the four species were frequently detected in soil-related samples in this study, suggesting that *Armatimonadota* bacteria may contribute to the growth of terrestrial plants as well as aquatic plants. This possibility is further supported by previous 16S rRNA gene amplicon sequencing studies reporting the widespread distribution of *Armatimonadota* in terrestrial plant environments, such as rhizosphere soils [22,26,27,48,49]. Indeed, inoculation of strain LA-C6 into *Arabidopsis thaliana*, a model terrestrial plant, was shown to promote its growth (Appendix A). Similarly, *Acinetobacter calcoaceticus* strain P23, a PGPB isolated from *L. aequinoctialis*, is also known to enhance the growth of both terrestrial plants (*Lactuca sativa*) and aquatic plants [14]. Furthermore, PGPB isolated from the aquatic plant *Hydrocotyle umbellata* has also been reported to promote the growth of not only duckweeds but also terrestrial crops, such as cucumber and sorghum [50]. Collectively, these findings suggest that PGPB within the phylum *Armatimonadota*, including strain LA-C6, may possess the capacity to enhance the growth of both aquatic and terrestrial plants.

To date, three classes have been validly described within the phylum *Armatimonadota* (*Armatimonadia*, *Fimbriimonadia*, and *Chthonomonadia*). In addition, eight isolates have been reported from diverse environments including reed, thermal soil, rhizosphere soil, bark, and duckweed [21,28,40,41,42,47,51,52]. However, very little is known about their functional characteristics. Genome-based analyses have predicted several functional capabilities in certain *Armatimonadota* bacteria, including nitrogen and sulfur reduction in marine sediments, iron oxidation and reduction in hot spring environments, catabolic reduction of nitrate to ammonia, and the potential supply of coenzymes to anammox bacteria [53,54,55,56]. However, these studies have been largely confined to anaerobic or high-temperature environments. To date, no reports including metagenome analyses have addressed interactions between *Armatimonadota* members and plants. The present study sheds light on a previously unexplored aspect of the functional capabilities of the phylum *Armatimonadota* by revealing their potential role in promoting plant growth. In recent years, several studies have utilized metagenomic analyses to identify bacterial lineages harboring genes associated with PGP traits [57,58,59]. Although gene-based screening is an efficient approach for discovering potential PGPB, it may fail to detect PGPB strains, such as strain LA-C6, that possess few or none of the canonical PGP genes. This study underscores the importance of isolating microorganisms from environmental samples, and such culture-based approaches are expected to facilitate the discovery of novel functional traits and ecological roles within underexplored bacterial lineages.

## 4. Materials and Methods

### 4.1. Bacterial Strains

In the previous study [21], we isolated a taxonomically novel bacterial strain belonging to the phylum *Armatimonadota* from fronds of *L. aequinoctialis*. The 16S rRNA gene of this strain was amplified by PCR using the primers Eub-8F (5′-AGAGTTTGATCMTGGCTCAG-3′) and Eub-1492R (5′-TACGGHTACCTTGTTACGACTT-3′) [60,61]. PCR products were purified using ExoSAP-IT Express PCR Product Cleanup Reagent (ThermoFisher Scientific, Waltham, MA, USA). Purified PCR products were sequenced using an Applied Biosystems^TM^ 3730*xl* DNA analyzer (ThermoFisher Scientific). The sequence was compared with those present in the EzBioCloud database (https://www.ezbiocloud.net/). A phylogenetic tree was constructed using the Neighbor-joining method with MEGA X software [62].

### 4.2. Plant Materials and Growth Conditions

Six species of duckweeds, *L. minor*, *L. aequinoctialis*, *S. polyrhiza*, *L. punctata*, *Wolffia arrhiza*, and *Wolffia globosa*, were used as plant materials. All duckweeds were aseptically cultivated in plant culture pots (420 mL, 70 mm diameter; IWAKI, Tokyo, Japan) containing 200 mL of modified Hoagland medium [17]. The pots were placed in a growth chamber at 25 °C under a light intensity of 15,000 lux with a 16–h light/8–h dark photoperiod.

### 4.3. Preparation of Bacterial Strains

All bacterial strains were grown on R2A agar plates. The colonies were swabbed and suspended in 200 mL of liquid R2A medium in 500 mL flasks. Each flask was incubated at 25 °C in the dark with shaking at 150 rpm. Cells in the late exponential phase were harvested by centrifugation at 15,000× *g* for 3 min. The cells were washed twice in sterile modified Hoagland medium and resuspended in 20 mL of the same medium. The suspensions were adjusted to an OD_660_ = 0.3 in 40 mL of modified Hoagland medium for co-culture with duckweeds.

### 4.4. Evaluation of the PGP Effect of Strain LA-C6 on Duckweeds Cultivated in Bacterial Suspension

Two fronds of each duckweed species were soaked and cultivated in modified Hoagland medium containing bacterial cells (final OD_660_ = 0.3) in a growth chamber (*n* = 5) (cell suspension method) (Appendix A). *L. minor*, *L. aequinoctialis*, and *L. punctata* were cultivated in plant culture tubes (150 mL, 40 mm diameter; IWAKI) containing 40 mL of modified Hoagland medium. *S. polyrhiza* was cultivated in plant culture pots containing 100 mL of modified Hoagland medium. *W. arrhiza* and *W. globosa* were cultivated in 12-well polystyrene microplates containing 7 mL of modified Hoagland medium supplemented with EDTA (5 mg/L). After 14 days of cultivation, the number of fronds was measured. As a negative control, each species of aseptic duckweed was cultivated in modified Hoagland medium in the absence of bacteria. Statistical differences were identified by a two-tailed Student’s *t*-test using Microsoft Excel, version 16.100.3.

### 4.5. Evaluation of the PGP Effect of Strain LA-C6 on Bacteria-Attached Duckweeds in Sterile Medium

As shown in Appendix A, five to ten plants of each duckweed species, *L. minor*, *L. aequinoctialis*, *S. polyrhiza*, *L. punctata*, *W. arrhiza*, and *W. globosa*, were placed in plant culture tubes filled with 40 mL of modified Hoagland medium (pre-attached method). Bacterial cells of strain LA-C6 were suspended in modified Hoagland medium in each tube (final OD_660_ = 0.3). To allow bacterial attachment, each duckweed plant was co-cultured with strain LA-C6 for 3 days. Then, one co-cultured plant of *L. minor*, *L. aequinoctialis*, or *L. punctata* was transferred into new plant culture tubes containing 40 mL of modified Hoagland medium. One co-cultured plant of *S. polyrhiza* was transferred into new plant culture pots containing 100 mL of modified Hoagland medium. Furthermore, two plants (four fronds) of co-cultured *W. arrhiza* and *W. globosa* were transferred into 12-well polystyrene microplates containing 7 mL of modified Hoagland medium supplemented with EDTA (5 mg/L). This process minimized carryover of nutrients from the original growth medium and of nutrient leakage from dead bacterial cells. All duckweed samples were cultivated in a growth chamber (*n* = 5). After 14 days of cultivation, the number of fronds was measured. As a negative control, aseptic plants of each duckweed species were cultivated in modified Hoagland medium without bacteria. Statistical differences were identified by a two-tailed Student’s *t*-test as stated above.

### 4.6. PGP Effect of Armatimonadota Strains on Duckweeds

The PGP effect of *Armatimonadota* bacteria belonging to phylogenetic lineages distinct from strain LA-C6 was examined. The bacterial strains used in this experiment are listed in Table 1. *A. rosea* strain YO-36^T^ and *C. corticalis* strain AX-7^T^ were purchased from the NITE Biological Resource Center (NBRC). *F. ginsengisoli* strain Gsoil 348^T^ was obtained from Japan Collection of Microorganisms (JCM). As a positive control, *Acinetobacter calcoaceticus* P23, a well-known PGPB for duckweeds [14], was also tested for this experiment. Preparation of bacterial inoculants was performed as described above (Appendix A). One *L. minor* plant (two fronds) was placed in a plant culture tube containing 40 mL of modified Hoagland medium. Bacterial cells of four *Armatimonadota* strains, including strain LA-C6, were inoculated into modified Hoagland medium in each tube (final OD_660_ = 0.3). All plants were co-cultured with the bacterial strains for 3 days. Then, one co-cultured *L. minor* plant was transferred into new plant culture tubes containing 40 mL of modified Hoagland medium. All test tubes were incubated in a growth chamber (*n* = 3). After 14 days of cultivation, the number of fronds and the dry weight were measured. Statistical testing was performed using a two-tailed Dunnett’s test with the Microsoft Excel, version 16.100.3.

### 4.7. Bacterial Colonization on Duckweed Surfaces

To evaluate bacterial colonization on duckweed surfaces, aseptic plants of *L. minor* and *L. aequinoctialis* were co-cultured with bacterial cells (OD_660_ = 0.3) in plant culture tubes containing 40 mL of modified Hoagland medium. All plant tubes were incubated for 14 days in a growth chamber (*n* = 3). During the incubation period, three duckweed plants were harvested from each tube on days 3, 7, 10, and 14. Each plant sample was gently washed twice with 10 mL of modified Hoagland medium in a 25 mL conical tube. After washing, the excess water was gently absorbed with a sterile paper towel. The fresh weight of each plant was measured, and the samples were stored at −20 °C.

### 4.8. DNA Extraction and qPCR Analysis

Genomic DNA of strain LA-C6 was extracted from duckweed samples using Cica Geneus DNA Extraction Reagent ST (Kanto Chemical Co., Tokyo, Japan) and subsequently purified with a Zymo-Spin Column I (Funakoshi, Tokyo, Japan) according to the manufacturer’s protocol. The purified DNA was quantified using SYBR Green-based real-time PCR (qPCR) with the primer set LA-C6 F1 (5′-TGAACAACGGCGTCGTGACG-3′) and LA-C6 R1 (5′-ATAGCTTGCACCACCCACGT-3′), targeting the region from 1718 bp to 1881 bp (164 bp fragment) of the RNA polymerase beta subunit (*rpoB*) gene in strain LA-C6. The thermal cycling conditions were as follows: 95 °C for 1 min, followed by 35 cycles of 94 °C for 15 s, 57 °C for 30 s, and 72 °C for 20 s. Ten-fold serial dilutions (1.67 × 10^3^ to 1.67 × 10^8^ copies/μL) of amplicons targeting the *rpo*B gene of strain LA-C6 (391 bp) were used as standards. This amplicon was generated using the universal primer set rpoB 1675 (5′-TGYCCGATYGAAACACCKGARGG-3′) and rpoB 2063 (5′-TGACGYTG CATGTTCGMNCCCAT-3′) [63], under the same thermal cycling conditions described in previous studies [64].

### 4.9. Microscopic Observations

Two aseptic duckweed species, *L. minor* and *L. aequinoctialis*, were co-cultured with strain LA-C6 (final OD_660_ = 0.3) in plant culture tubes containing 40 mL of modified Hoagland medium for 7 days. To visualize bacterial cells attached to the plants, each plant was stained using the LIVE/DEAD BacLight Bacterial Viability Kit for microscopy (ThermoFisher Scientific). The stained plants were mounted on glass slides and overlaid with coverslips. Images were obtained using a fluorescence microscope (Axio Observer.Z1, Zeiss, Jena, Germany) with the software AxioVision 4.9.1.0 (Zeiss). 

### 4.10. Assays for Plant Growth-Promoting Traits

PGP traits (siderophore production, phosphate solubilization, and IAA production) of strain LA-C6 and other *Armatimonadota* bacteria were examined using standard methods. Siderophore production was tested using chrome azurol S (CAS) agar (1.5% *w*/*v*) as previously described [65]. Phosphate solubilization was measured using Pikovskaya’s agar (1.5% *w*/*v*) as previously described [66]. Colonies of each bacterial strain grown on R2A agar were streaked onto the respective assay plates using inoculating loops. IAA production was evaluated using the Salkowski reagent in R2A liquid medium supplemented with 0.5% (*w*/*v*) L-tryptophan, following a previously described method [67].

### 4.11. Genome Analysis of a Taxonomically Novel Armatimonadota Strain LA-C6

The genomic DNA of strain LA-C6 was extracted by chemical and enzymatic procedures, as described previously [68]. In brief, lysozyme, achromopeptidase, and proteinase K were used to lyse cells, followed by genomic DNA purification using the phenol–chloroform method. The concentration of genomic DNA was determined using a QuantiFluor dsDNA System with a Quantus Fluorometer (Promega, Madison, WA, USA). Library preparation and sequencing were performed using the Revio Polymerase kit and the Revio sequencing system (PacBio, Menlo Park, CA, USA). Assembly of long reads was carried out using Flye (ver. 2.9.2-b1786) with default parameters. Gene prediction was performed using Prokka (ver. 1.14.6) [69] and BlastKOALA KEGG service (version 3.0) [70]. Prediction of metabolic pathways and enzymes was performed using NCBI BLASTp (https://blast.ncbi.nlm.nih.gov/Blast.cgi?PAGE=Proteins accessed on 13 February 2024) and Pfam [71].

### 4.12. IMNGS-Based 16S rRNA Amplicon Survey in Natural Environments

To identify the relative abundance of strain LA-C6 and other *Armatimonadota* strains in environments, the publicly available 16S rRNA amplicon datasets were surveyed using the IMNGS platform [72]. Samples were selected based on their environmental origins, including plant-associated, freshwater, and soil environments. Full-length or nearly full-length 16S rRNA gene sequences of strain LA-C6, *A. rosea* strain YO-36^T^, *C. corticalis* strain AX-7^T^, and *F. ginsengisoli* Gsoil strain 348^T^ were used as query sequences for similarity searches against the selected datasets, using 97% sequence identity cutoffs and a minimum alignment length of 200 bp. In the selected environments, target 16S rRNA sequences detected at >0% relative abundance were recorded.

### 4.13. PGP Effect of Strain LA-C6 on Terrestrial Plants

To examine the PGP effect of strain LA-C6 on terrestrial plants, *Arabidopsis thaliana* ecotype Columbia (Col-0) was used for the PGP test. Surface sterilization of seeds was carried out according to a previously described method [73]. The sterilized seeds were germinated on half-strength Murashige and Skoog (1/2 MS) agar plates in the dark at 4 °C for 3 days. After germination, ten seedlings were transferred to new 1/2 MS agar plates and incubated for 5 days. The plates were kept in a growth chamber at 22 °C under a 12–h light/12–h dark photoperiod at a light intensity of 15,000 lux. After five days, seven seedlings were placed on each new 1/2 MS agar plate, and 2.5 µL of strain LA-C6 suspension (in sterile distilled water; final OD_660_ = 1.0) was inoculated into each seedling. The inoculated plants were cultivated for an additional 26 days under the same conditions. Non-inoculated plants were also grown under the same conditions and used as negative controls. After cultivation, the PGP effect of strain LA-C6 was evaluated based on the fresh weight and root length of the plants. Statistical differences were identified by a two-tailed Student’s *t*-test as stated above.

## Figures and Tables

**Figure 1 ijms-26-09824-f001:**
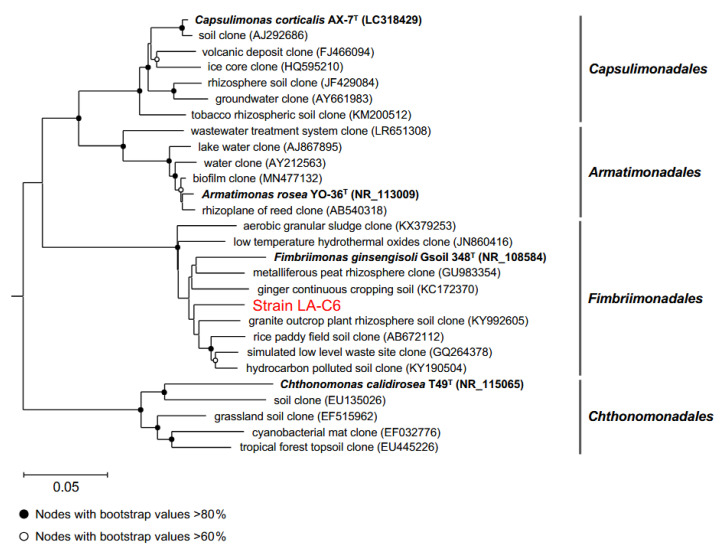
Phylogenetic tree of the novel isolate, related *Armatimonadota* type species, and environmental sequences based on the 16S rRNA gene. The tree was constructed by the Neighbor-joining method. Type species and environmental clones of *Abditibacteriota* were used as outgroups (accession numbers: NR_179096, JQ769997, JN178695, JF024887, AB374370). Validly described strains are shown in bold. Bootstrap values were calculated from 1000 replicates. Nodes with bootstrap values >80% and >60% are shown as filled and open circles, respectively. Scale bar: 0.05 nucleotide substitutions per site.

**Figure 2 ijms-26-09824-f002:**
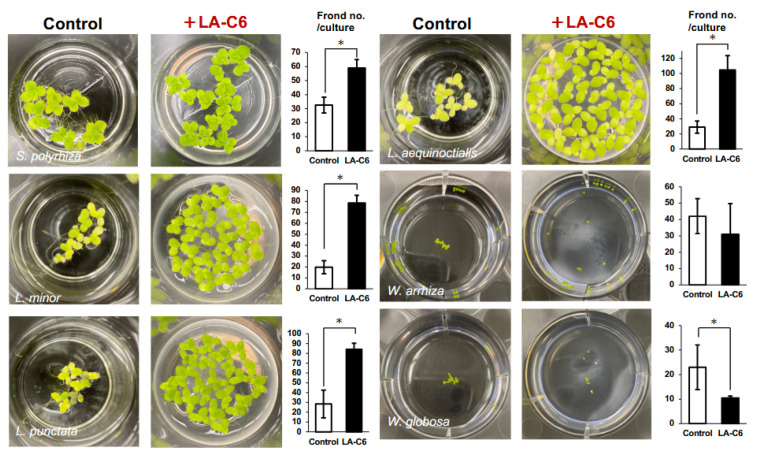
Plant growth-promoting effects of strain LA-C6 on six duckweed species when each plant was soaked in modified Hoagland medium containing cultured bacterial cells (cell suspension method). Images were taken on day 14 of incubation. The number of fronds was counted on day 14. *n* = 5; *, *p* < 0.05; bars, error bars represent SE.

**Figure 3 ijms-26-09824-f003:**
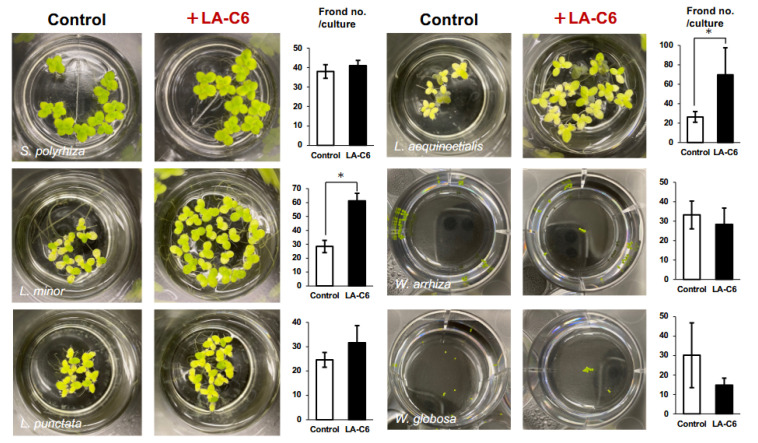
Plant growth-promoting effects of strain LA-C6 on six duckweed species when bacteria-attached duckweeds were cultivated in new tubes containing sterile modified Hoagland medium (pre-attached method). Images were taken on day 14 of incubation. The number of fronds was counted on day 14. *n* = 5; *, *p* < 0.05; bars, error bars represent SE.

**Figure 4 ijms-26-09824-f004:**
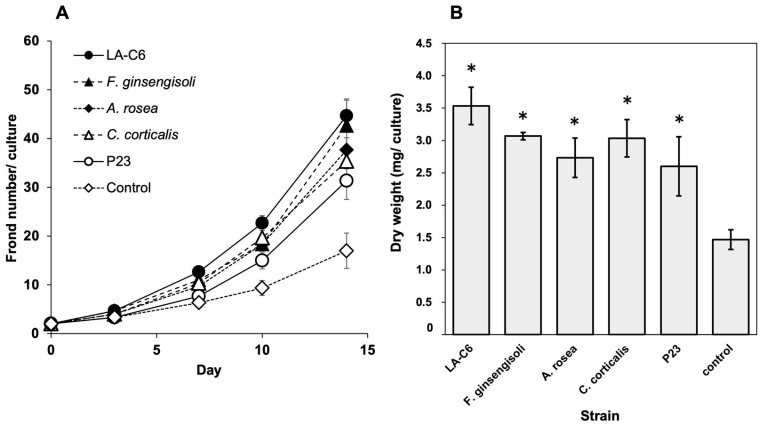
Plant growth-promoting effects of three *Armatimonadota* strains (representing different species) and strain LA-C6 on *L. minor*. (**A**) The number of fronds of *L. minor* and (**B**) the dry weight of *L. minor* after 14 days of cultivation. *n* = 3; *, *p* < 0.05; error bars represent SE.

**Figure 5 ijms-26-09824-f005:**
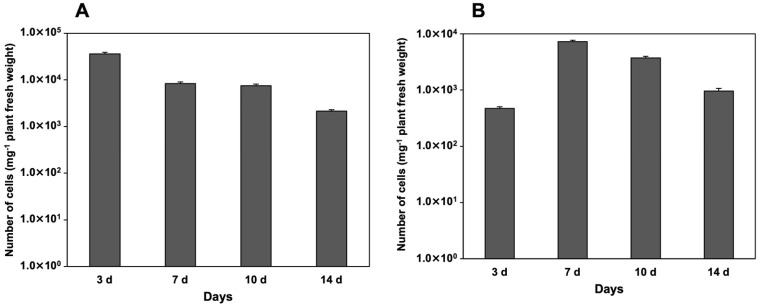
Cell number of strain LA-C6 on the fronds and roots of (**A**) *L. minor* and (**B**) *L. aequinoctialis* during 14 days of cultivation. *n* = 3; error bars represent SE.

**Figure 6 ijms-26-09824-f006:**
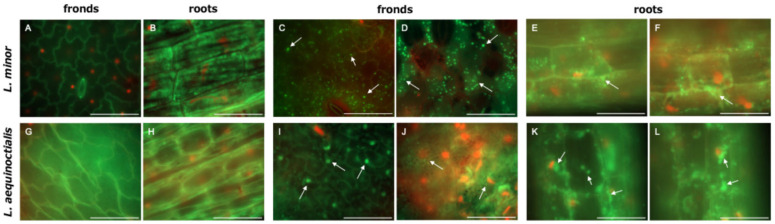
Fluorescence micrographs of live/dead-stained fronds and roots of aseptic (**A**,**B**) *L. minor* and (**G**,**H**) *L. aequinoctialis*, and of strain LA-C6 co-cultured (**C**,**D**,**E**,**F**) *L. minor* and (**I**,**J**,**K**,**L**) *L. aequinoctialis.* Live and dead cells are visualized in green and red, respectively. Arrows indicate live stained cells. Scale bars, 50 μm.

**Figure 7 ijms-26-09824-f007:**
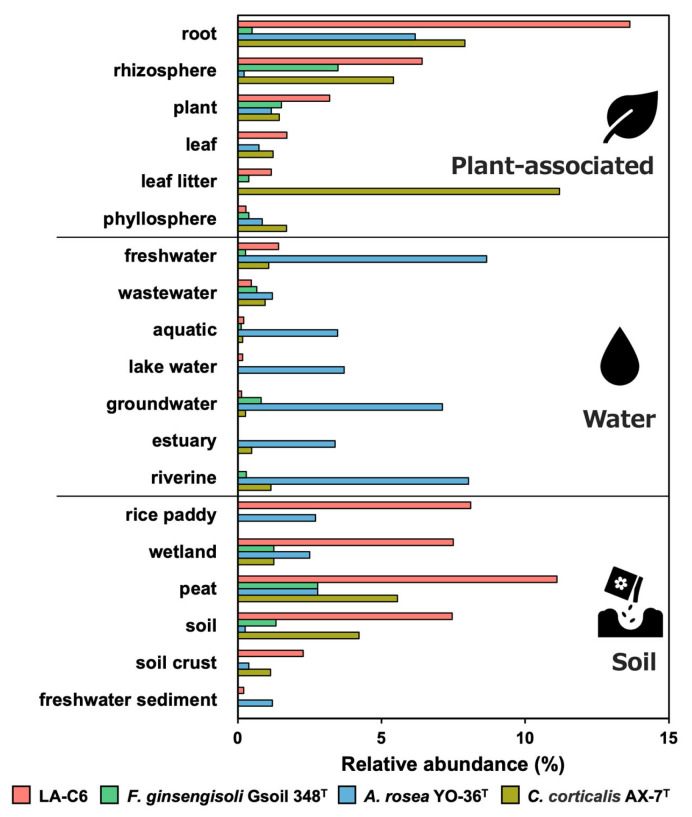
Distributions of strain LA-C6 and other *Armatimonadota* strains in different environmental samples. Bars indicate the frequency of detection (relative abundance >0%) for each species in three sample types (plant-associated, water, and soil).

**Table 1 ijms-26-09824-t001:** List of *Armatimonadota* strains evaluated for their plant growth–promoting effect on *L. minor*.

Strain	Order	Isolation Source
*A. rosea* strain YO-36^T^	*Armatimonadales*	rhizoplanes of reeds
*F. ginsengisoli* strain Gsoil 348^T^	*Fimbriimonadales*	ginseng field soil
*C. corticalis* strain AX-7^T^	*Capsulimonadales*	trunk surface of Japanese beech
Strain LA-C6	*Fimbriimonadales*	fronds of duckweeds

## Data Availability

The complete genome of strain LA-C6 has been submitted in GenBank under accession number AP038796. The raw sequencing data have been deposited in the SRA under accession number DRR621043. The BioSample and BioProject accession numbers are SAMD00842066 and PRJDB19541, respectively.

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
