# Peer review of "Diverse Members of the Phylum Armatimonadota Promote the Growth of Aquatic Plants, Duckweeds"

_ijms, 2025, doi:10.3390/ijms26199824_

Round 1

Reviewer 1 Report

Comments and Suggestions for Authors

General comment: This manuscript reports the plant growth-promoting effects of members of the phylum Armatimonadota on duckweed species and Arabidopsis thaliana. The study brings novelty showing evidences that a rarely cultivated bacterial group can significantly stimulate plant growth. The experiments are well designed, data are clearly presented, and the findings are discussed in the context of previous literature, however the number of replicates is low which can limit robustness of the results. The manuscript is well written with quality English, however there a few long sentences that should be revised to improve fluidity.

Comment 1: There is a typing error on the word “Figure”  in the legend of the all the figures, it is incorrectly written “Figureure”.

Comment 2: In the Abstract the sentence “In this study, we report the plant growth-promoting effects of members of the phylum Armatimonadota, which are distributed in natural environments but whose ecological roles remain poorly understood” could be more fluid if replaced by “widely distributed… but with poorly understood roles” (lines 21-23)

Comment 3: How can you exclude the possibility that the observed plant growth promotion is due to nutrient or metabolite carryover rather than active bacterial colonization?

Comment 4: Wolffia species did not respond to LA-C6 inoculation, could this be due to root absence limiting bacterial colonization and nutrient exchange?

Comment 5: In the discussion some sentences are too long, shorten sentences can improve readability.  

Comment 6: When you note that IAA production likely does not explain the effect, it might be useful further discuss possible alternative mechanisms.

Author Response

Comment 1: There is a typing error on the word “Figure”  in the legend of the all the figures, it is incorrectly written “Figureure”.

>Thank you for pointing out the typographical error. We have corrected all instances of “Figureure” to “Figure” in the figure legends and in the main text.

Comment 2: In the Abstract the sentence “In this study, we report the plant growth-promoting effects of members of the phylum Armatimonadota, which are distributed in natural environments but whose ecological roles remain poorly understood” could be more fluid if replaced by “widely distributed… but with poorly understood roles” (lines 21-23)

>We appreciate this suggestion. However, in order to address Reviewer 2’s concern about the inconsistency in whether the manuscript focuses on a single strain or multiple strains, we revised the Abstract to clarify this point. Consequently, we have removed this sentence from the Abstract.

Comment 3: How can you exclude the possibility that the observed plant growth promotion is due to nutrient or metabolite carryover rather than active bacterial colonization?

>The results obtained using the pre-attached method (Fig. 3) indicate that nutrient or metabolite carryover is unlikely to explain the observed plant growth promotion. In this method, duckweeds pre-incubated with bacteria were transferred to sterile modified Hoagland medium, thereby minimizing the influence of nutrients derived from the original growth medium as well as any nutrient leakage from dead bacterial cells. Despite this, strain LA-C6 consistently promoted growth across a wide range of duckweed species, supporting the conclusion that the observed effects are due to active bacterial colonization. To make this point clearer, we have also added the following sentence to the Materials and Methods section:

“This process minimized carryover of nutrients from the original growth medium and of nutrient leakage from dead bacterial cells” (L387–L388 in the revised manuscript).

Comment 4Wolffia species did not respond to LA-C6 inoculation, could this be due to root absence limiting bacterial colonization and nutrient exchange?

>Thank you for your insightful comment. As you pointed out, the absence of roots in Wolffia may limit bacterial colonization and nutrient exchange, which could explain the lack of growth-promoting effects were not observed. In addition, a recent study (doi: 10.1186/s40793-025-00759-6) suggests that microbial communities differ between Wolffia and rooted duckweed species. These findings support the possibility that strain LA-C6 is better adapted to colonizing rooted duckweeds, but may be less suited to rootless duckweeds such as Wolffia.

Comment 5: In the discussion some sentences are too long, shorten sentences can improve readability.  

>We agree that some sentences in the Discussion section were too long and affected readability. We have revised these sentences to make them shorter and more concise (L230–L233, L269–L272, and L314–L317 in the revised manuscript).

Comment 6: When you note that IAA production likely does not explain the effect, it might be useful further discuss possible alternative mechanisms.

>We have added a discussion of possible alternative mechanisms that may contribute to the plant growth-promoting effect of strain LA-C6 as follows:

“Therefore, alternative mechanisms may be involved, such as the modulation of duckweed stress tolerance (e.g., oxidative stress, light intensity, and nutrient deficiency), or the supply of unknown substances that enhance metabolism and biomass yield. However, these possibilities remain to be further clarified.” (L287–L291 in the revised manuscript).

Reviewer 2 Report

Comments and Suggestions for Authors

The manuscript "Diverse members of the phylum Armatimonadota promote the growth of aquatic plants, duckweeds" is well written. One major concern is why the authors did not identify the isolates to the species level. They have conducted both 16S rRNA gene sequencing and complete genome sequencing, yet the genus and species remain undetermined. Is the 16S rRNA gene accession number available? If so, given the low percentage identity, should this be considered a novel species? It is unclear whether the study is focused on a single strain or multiple strains. The manuscript refers to "diverse members," but at some points only one strain is discussed, while in other places three strains are mentioned. This inconsistency needs clarification.

Author Response

One major concern is why the authors did not identify the isolates to the species level. They have conducted both 16S rRNA gene sequencing and complete genome sequencing, yet the genus and species remain undetermined. Is the 16S rRNA gene accession number available? If so, given the low percentage identity, should this be considered a novel species?

>Thank you for your valuable comment. Based on homology analysis of the 16S rRNA gene and phylogenetic tree within the phylum Armatimonadota, strain LA-C6 is suggested to represent a novel bacterial lineage at the genus level rather than the species level. The partial 16S rRNA gene sequence has been deposited in GenBank under accession number LC523985 (L88–L89 in the revised manuscript), and the full-length 16S rRNA sequence is available from the complete genome registered in this study (accession number AP038796) (L91–L92 in the revised manuscript).

Although the 16S rRNA gene identity to the closest described species is below the threshold typically used for species delineation, formal recognition as a novel species or genus requires further comprehensive phenotypic and chemotaxonomic characterization, which is beyond the scope of this study. Therefore, we refer to the isolate as Fimbriimonadaceae bacterium strain LA-C6 throughout the manuscript. To clarify this point, we have added the sentence as follows:

“Based on 16S rRNA gene analysis, this strain represents a novel genus-level lineage, and is referred to as Fimbriimonadaceae bacterium strain LA-C6.” (L22–L24 in the revised manuscript)

“These results clearly indicate that this strain represents a phylogenetically novel lineage at least at the genus level, therefore it is referred to as Fimbriimonadaceae bacterium strain LA-C6.” (L98–L100 in the revised manuscript).

It is unclear whether the study is focused on a single strain or multiple strains. The manuscript refers to "diverse members," but at some points only one strain is discussed, while in other places three strains are mentioned. This inconsistency needs clarification.

>Thank you for pointing out this inconsistency. We have clarified this issue primarily in the Abstract and the Introduction sections. In particular, we have stated that the main focus of this study is the discovery of plant growth-promoting (PGP) effects of strain LA-C6, and we additionally found that three other known Armatimonadota species also exhibited PGP traits (L21–L22, L27–L30, L30–L31, and L79–L80 in the revised manuscript). If further clarification is necessary, we are also open to revising the title from “Diverse members of the phylum Armatimonadota promote the growth of aquatic plants, duckweeds” to “A taxonomically novel Armatimonadota strain promotes the growth of aquatic plants, duckweeds” to better reflect the study’s focus.

Round 2

Reviewer 2 Report

Comments and Suggestions for Authors

Thanks for the clarification and addressing all comments, I have no further comments.